# US Evaluation of Topical Hemostatic Agents in Post-Thyroidectomy

**DOI:** 10.3390/cancers15092644

**Published:** 2023-05-07

**Authors:** Vincenzo Dolcetti, Eleonora Lori, Daniele Fresilli, Giovanni Del Gaudio, Chiara Di Bella, Patrizia Pacini, Vito D’Andrea, Fabrizio Maria Frattaroli, Giulia Giordana Vallone, Piero Liberatore, Daniele Pironi, Gian Luigi Canu, Pietro Giorgio Calò, Vito Cantisani, Salvatore Sorrenti

**Affiliations:** 1Department of Radiological, Anatomo-Pathological Sciences, “Sapienza” University of Rome, Viale Regina Elena 324, 00161 Rome, Italy; daniele.fresilli@hotmail.it (D.F.); g.d.gaudio@gmail.com (G.D.G.); chiara.dibella@uniroma1.it (C.D.B.); patry.shepsut91@gmail.com (P.P.); vito.cantisani@uniroma1.it (V.C.); 2Department of Surgery, “Sapienza” University of Rome, 00161 Rome, Italy; eleonora.lori@uniroma1.it (E.L.); vito.dandrea@uniroma1.it (V.D.); daniele.pironi@uniroma1.it (D.P.);; 3Department of Surgery “P. Stefanini”, Faculty of Medicine, “Sapienza” University of Rome, Piazzale Aldo Moro 5, 00185 Rome, Italy; fabrizio.frattaroli@uniroma1.it (F.M.F.); giuliagiordana.vallone@uniroma1.it (G.G.V.); piero_liberatore@hotmail.com (P.L.); 4Department of Surgical Sciences, University of Cagliari, 09042 Monserrato (Cagliari), Italy; pgcalo@unica.it (P.G.C.); gianlu_5@hotmail.it (G.L.C.)

**Keywords:** topical hemostatic agents, Oxitamp^®^, Tisseel^®^, thyroidectomy, ultrasound, radiological follow-up

## Abstract

**Simple Summary:**

To reduce bleeding-related morbidity in th yroid surgery, new techniques and devices for controlling bleeding have been introduced in addition to traditional hemostasis. Innovative technological tools based on mechanical hemostasis systems, conventional mono- and bipolar electrosurgery systems, and radiofrequency and ultrasound systems can be variably associated with topical hemostatic agents and surgical sealants to improve hemostasis and avoid intra- and post-operative bleeding. The aim of the study was to describe the ultrasound (US) appearance of topical hemostatics in patients who were undergoing thyroidectomy. In particular, the study focused on the ultrasound appearance of a hemostatic swab, which in cases of incomplete resorption, can simulate disease recurrence (in oncological patients) or native gland residue. The study evaluated the possible advantages and disadvantages of the application of these agents to thyroid surgery, in particular those based on oxidized and regenerated cellulose and fibrin glue. The preliminary data reported in this study show that the effectiveness of hemostatic fibrin glue in preventing bleeding is comparable to oxidized regenerated cellulose swab; however, unlike the swab, the fibrin glue does not show any detectable residues at the US examination.

**Abstract:**

Background: the aim of this study was to describe the ultrasound appearance of topical hemostatics after thyroidectomy. Methods: we enrolled 84 patients who were undergoing thyroid surgery and were treated with two types of topical hemostats, 49 with an absorbable hemostat of oxidized regenerated cellulose (Oxitamp^®^) and 35 with a fibrin glue-based hemostat (Tisseel^®^). All patients were examined using B-mode ultrasound. Results: In 39 patients of the first group (approximately 80%), a hemostatic residue was detected and in some cases confused with a native gland residue, or with cancer recurrence in oncological patients. No residue was detected in patients in the second group. The main ultrasound characteristics of the tampon were analyzed and arranged according to predefined patterns, and suggestions to recognize it and avoid wrong diagnoses were provided. A part of the group of patients with tampon residue was re-evaluated after 6–12 months, ensuring that the swab remained for months after the maximum resorption time declared by the manufacturer. Conclusions: with equal hemostatic effectiveness, the fibrin glue pad is more favorable in the ultrasound follow-up because it creates reduced surgical outcomes. It is also important to know and recognize the ultrasound characteristics of oxidized cellulose-based hemostats in order to reduce the number of diagnostic errors and inappropriate diagnostic investigations.

## 1. Introduction

Thyroid disorders are very common in the general population. Thyroid dysfunction, hyperthyroidism and hypothyroidism are present in 6% of the United States population. Benign nodules, also known as nodular goiter, are very common, with a prevalence of up to 68% [1]. The thyroid cancer is the most frequent of the endocrine neoplasms. More common in women than men, thyroid cancer is the fifth most common malignancy in the United State population and its incidence is increasing worldwide due to improved diagnostic imaging and more accurate cytological classification [2,3,4,5]. Both benign and malignant thyroid disorders often require surgical treatment. More than 100,000 thyroidectomies are performed each year in the United States [1]. The surgical management of thyroid disease, whether benign or malignant, has evolved over the years, trying to reduce the complications associated with this surgery. The incidence of thyroid surgery morbidity has been significantly reduced with the standardization of the capsular dissection technique, which has been in use since 1973, and total thyroidectomy for the treatment of benign and malignant thyroid disease has become increasingly popular [6]. Currently, the most significant and frequent complications associated with thyroid surgery are hypoparathyroidism and recurrent laryngeal nerve injury. However, although rare, post-operative bleeding is a life-threatening complication, which can occur in up to 2% of cases [7,8,9].

Hemostasis is an important issue in thyroid surgery because bleeding complications still have the highest incidence rate in both the intra- and post-operative period [10,11]. The criticality of bleeding, whether intraoperative or post-operative, is caused by the obscuring of important anatomical structures, making surgical dissection more complicated. Bleeding, in fact, can indirectly cause injury to the recurrent laryngeal nerves or parathyroid glands due to the execution of blind maneuvers [12]. To reduce bleeding-related morbidity, the use of appropriate hemostatic surgical techniques is essential. With the advent of modern surgery, in addition to traditional hemostasis, which is performed by tying vessels with sutures or with metal clips or absorbable materials, new techniques and devices for controlling bleeding have been introduced. These new surgical techniques adopt innovative technological tools based on mechanical hemostasis systems, conventional mono- and bipolar electrosurgery systems, and radiofrequency and ultrasound systems [13,14,15].

Topical hemostatic agents and surgical sealants play a complementary role to the surgical hemostasis techniques. In recent years, numerous topical hemostatic agents that are useful for facilitating hemostasis in cases in which traditional methods (mechanical, thermal and chemical methods) are ineffective or impracticable have been developed. Bleeding in thyroid surgery occurs frequently due to contiguity with anatomical areas in which electrocauterization or high-frequency vibration could be harmful to sensitive structures, such as the recurrent laryngeal nerve [16]. Topical hemostatics and surgical sealants can also be further divided into animal derivatives, human derivatives, plant derivatives and synthetics, according to the nature of the material. They can also be distinguished according to the mechanism of action [17].

The different topical hemostasis techniques represent an important topic for ultrasound follow-up in patients after thyroidectomy. In particular, one of them attracted our attention: the absorbable hemostat of oxidized regenerated cellulose with pH acid called Oxitamp^®^. The oxidized cellulose applied at the active bleeding site generates a “sponge” effect that activates the coagulative cascade that transforms itself into a gelatinous mass that traps blood proteins, platelets and red blood cells [18,19]. Since it is a sterile material that undergoes spontaneous resorption without causing an inflammatory reaction, Oxitamp^®^ can be intentionally left at the operating site [18]. Specifically, the manufacturer reports that the average full reabsorption time of the product is 5–8 days [19], while studies available in the literature on similar products based on oxidized and regenerated cellulose indicate that they have a complete reabsorption time of 14 days [20]. However, in our clinical experience, we found, during the ultrasound evaluation, that the average length of tampon persistence at the site of hemostasis is much higher than that reported in the data sheet. This leads to clinical and diagnostic errors and therefore to unnecessary in-depth investigations and the lengthening of healing times.

The aim of the study was to identify the surgical outcomes of the Oxitamp^®^ treatment, avoid pitfalls during follow-up, and to compare it with a fibrin glue-based hemostatic agent called Tisseel^®^.

## 2. Materials and Methods

This study was divided into two phases. In the first phase, we set three main objectives: to identify and describe the ultrasound characteristics of the residue of oxidized regenerated cellulose; to verify how many patients the residue was observed in at a variable time distance from the thyroid surgery; to establish the existence of a possible temporal relation between its morpho-dimensional characteristics and the time elapsed since surgery. In this way, we properly identified the surgical outcomes of the Oxitamp^®^ treatment, avoiding pitfalls during follow-up.

In the second phase of the study, the group of patients treated with Oxitamp^®^ was compared with a second group of patients undergoing thyroid surgery but treated instead with a local fibrin glue-based hemostatic agent called Tisseel^®^.

The study included 84 patients who were undergoing hemithyroidectomy or total thyroidectomy between May 2020 and January 2023. Among these 84 patients, 36 were men and 48 were women (this difference is probably due to the different incidence levels of thyroid nodule between the two sexes, which is higher in women), with an age range between 20 and 87 years (mean age 56.2). The two groups of patients were homogeneous in terms of gender and age and in terms of the diagnosis they had received (oncological pathology vs. surgical pathology, such as euthyroid multinodular goitre). In order to ensure the maximum homogeneity of the sample, patients who were undergoing total thyroidectomy and lymphadenectomy of the central or laterocervical compartment were excluded from the study. Demographic and clinical data are summarized in Table 1.

All patients were operated on in an elective surgery and were in a good general condition and had euthyroidism. None of the patients examined suffered from coagulation disorders and all patients had an INR within the reference values (0.9–1.1). Any anticoagulant therapies were suspended 5 days before the procedure and replaced with low-molecular-weight heparin (LMWH) at a therapeutic dose, with the last administration 12 h before the operation and the resumption of heparin therapy 24 h after it.

All procedures were performed under general anesthesia, following a preoperative anesthetic evaluation of the patient, the day before and the morning of the operation.

All operations were performed by the same surgeon, who has extensive experience in thyroid surgery.

In all procedures, an accurate hemostasis and vasal synthesis was routinely performed with conventional monopolar and bipolar electrosurgery techniques, which are associated with the use of a radiofrequency coagulator. In combination with these treatments, some patients were treated via the application of a topical hemostatic agent based on oxidized and regenerated cellulose (Oxitamp^®^) to the residual thyroidal space, while other patients were treated via the application of a human fibrin glue-based topical hemostatic (Tisseel^®^) to the residual thyroidal space. In particular, 49 patients were treated with Oxitamp^®^ and 35 were treated with Tisseel^®^. All patients underwent a baseline ultrasound examination using Color Doppler, which was carried out with 3–12 MHz multifrequency linear probes on a Samsung RS85 sonograph or with 5–14 MHz multifrequency linear probes on a Toshiba Aplio i800 sonograph. All the scans were performed by a radiologist with more than twenty years of experience in ultrasound and, in particular, the field of thyroid pathology. An evaluation was made between 3 and 24 months after surgery. The ideal time for the first post-operative scan was set between 3 and 9 months. Although this interval was respected for most patients, in some of those treated with Oxitamp^®^, it was considerably lengthened due to circumstances external to this study. In particular, eleven patients were enrolled in the study for more than 12 months after surgery because they had undergone previous diagnostic investigation, such as strict ultrasound surveillance (7 patients), MRI (1 patient) and scintigraphy (3 patients); one of them had undergone a needle aspiration evaluation. This delay, however, ensured a wide stratification of patients over time.

Post-surgical changes and the presence or absence of a residual swab at the level of the thyroid lodge were assessed in each patient. Then, a regression analysis was performed for positive cases, correlating the size of the residual swab with the time elapsed since surgery.

The comparison between the group treated with Oxitamp^®^ and the group treated with Tisseel^®^ was performed using the chi-square test for the discrete variables, and using the Student’s *t* test for the continuous variables. A *p*-value less than 0.05 was considered statistically significant.

Number Cruncher Statistical System (NCSS) and Microsoft Excel were the software used for all analyses.

## 3. Results

In the first phase of the study, 49 patients treated with Oxitamp^®^ were analyzed. In 39 of them (approximately 80%), the presence of an ultrasonographically viewable residue was found, while no residue of the swab material was displayed in the remaining 10 cases (approximately 20%), demonstrating its complete resorption.

This material has shown some specific ultrasonographically recognizable features, which are explained in detail in the next paragraph.

### 3.1. Ultrasound Features of Oxidized Regenerated Cellulose

The ultrasound appearance of the oxidized regenerated cellulose pad used in thyroid surgery has not been described in the literature. Although some scientific studies have reported that the oxidized regenerated cellulose pad possesses a wide variety of ultrasound characteristics, in our study, we found a less variable and more easily recognizable ultrasound appearance.

The ultrasound features of oxidized regenerated cellulose are summarized in the table below (Table 2).

Different features are usually mixed in the ultrasound swab evaluation, but overall, two distinct main patterns were found:A markedly hypoechoic pattern with well-defined margins, often with evidence of a peripheral halo (pseudo-capsule appearance), was found in 13 patients, which is approximately 34% of patients with evidence of residual swab (Figure 1a,b).A mildly hypoechoic or isoechoic pattern with ill-defined margins was found in 23 patients, which is approximately 58% of patients with evidence of residual swab (Figure 2a,b).

The remaining 3 patients (8% of patients with evidence of residual swab) showed different patterns, with different combinations of margins and ecogenicity. In our experience, we have also found different swab shapes, but the most common are ovoid or pseudo-ovoid shapes; these are the most frequent and most “thyroid-like” shape, and are caused when the remnant has completely obliterated the thyroid space that reproduces the native thyroid gland shape. This form is often associated with the mildly hypoechoic pattern with ill-defined margins (Figure 3a,b).

We also found a higher incidence of the first pattern in patients that were operated on for at least 18 months and a higher incidence of the second pattern in the earlier post-operative phases. However, this is not true for all patients, as demonstrated by the cases in which we found the second pattern even approximately three years after surgery.

### 3.2. Evaluation of the Residual Volume

The residual swab volume was also calculated during the ultrasound examination.

The results were stratified considering the time between the surgery and the ultrasound evaluation. Then, a regression analysis was carried out involving all 49 patients treated with Oxitamp^®^; this identified a weak and not statistically significant correlation between time and the residual volume of oxidized regenerated cellulose. The time–residual volume evaluation was also extended to 9 of the 39 patients who had undergone distant ultrasound re-evaluation (in periods ranging between approximately 3 months and 21 months): no patient showed the complete disappearance of the residue; in one third of the patients (six), there was an ultrasound-detectable reduction in size; and in the remaining third of patients (three), no significant change in size was detected. The regression analysis also showed results consistent with the previous ones, showing a weak and not statistically significant correlation between the time elapsed since the intervention and the size of the residue. The data are summarized in the chart below (Figure 4).

An important limitation of this type of analysis is represented by the fact that it was not possible to trace the precise quantity of swab used for hemostasis, which varied from patient to patient depending on the volume of the surgical cavity following thyroidectomy or hemithyroidectomy.

### 3.3. Comparison between Oxidized Regenerated Cellulose and Fibrin Glue

In the second phase of the study, the group of 35 patients treated with the fibrin glue hemostatic (Tisseel^®^) underwent a 6- or 12-month post-surgical ultrasound evaluation (an average delay of approximately 3 months was observed, in comparison with the group of patients treated with Oxitamp^®^). A solid material that was frankly identifiable as a hemostatic residue was found in none of the 35 patients analyzed, although in 5 patients there was a doubtful finding; this was described in the original report as “fibrotic tissue following surgery”. The difference between the two samples of patients has been traced back to the different mechanisms of action in hemostatics, which in the case of Tisseel^®^, does not involve the use and release of a solid buffer material.

### 3.4. Assessment of Post-Operative Blood Loss in Patients Treated with Oxidized Regenerated Cellulose in Comparison with Patients Treated with Fibrin Glue

At the end of each surgery, a suction drainage was positioned with the distal end in the residual thyroid area in order to objectively measure the post-operative blood loss. The drain was then removed in most cases at the beginning of the third post-operative day. A comparison of the drainage volume between the two groups was performed on the first and second post-operative day. On the first post-operative day, a mean drainage volume of 45.19 ± 23.84 mL of blood was found in the Oxitamp^®^-treated group, and a mean drainage volume of 35.94 ± 20.43 mL was found in the Tisseel^®^-treated group; comparing the mean blood volume values of the groups, a *p*-value of 0.1516 is obtained. Therefore, the difference between the means of the blood loss in the two groups is not statistically significant.

Considering the second post-operative day, a mean drain volume of 35.32 ± 23.26 mL of blood was found in the Oxitamp^®^-treated group and a mean drain volume of 41.88 ± 20.48 mL was found in the Tisseel^®^-treated group; comparing the mean blood volume values of the groups, a *p* value of 0.2983 is obtained. Therefore, even on the second post-operative day, there is no statistically significant difference in the means of the two groups.

The incidence of post-operative hemorrhage was then analyzed in the two groups of patients (one case with Oxitamp^®^ and one case with Tisseel^®^). No significant differences were found in terms of post-operative bleeding between the two groups.

## 4. Discussion

Post-operative hemorrhage is a rare but potentially fatal complication in thyroid surgery because a rapidly expanding hematoma in the thyroid lodge can cause severe airway compression, resulting in asphyxia. Sometimes, this may even require emergency tracheostomy as the patient’s intubation may be hindered by tracheal compression [6]. The onset of bleeding typically occurs within the first 24 h of surgery [21]. The main risk factors include pre-existing conditions such as hypertension, clotting disorders, anticoagulant therapies, Graves’ disease or other forms of hyperthyroidisms. Therefore, a meticulous intraoperative hemostasis is essential in order to prevent episodes of post-operative bleeding. The rich vascularization of the thyroidal parenchyma predisposes patients, however, to bleeding. It is therefore essential to perform a careful extracapsular dissection to avoid traumatizing patients.

In recent years, new facilities have been constructed to support traditional surgical techniques and ensure accurate hemostasis. Particularly interesting are the topical hemostatic agents that promote hemostasis and tissue sealing once applied directly to the hemorrhagic area. The purpose of our study was to evaluate the possible advantages and disadvantages of the application of these agents during thyroid surgery, and particularly those based on oxidized and regenerated cellulose and fibrin glue. They have proven to be very effective in reducing the volume of drained blood material in the first post-operative day compared with the exclusive use of traditional hemostasis techniques. We also have seen that these agents do not lead to statistically significant differences in the volumes of drained material after the first and the second post- operative days. Moreover, it should be noticed that neither of the two agents was superior to the other in reducing blood loss, even if the sample of patients treated with human fibrin glue was smaller than that of patients treated with oxidized and regenerated cellulose (only 35 Patients treated with Tisseel^®^ versus the 49 treated with Oxitamp^®^).

Data in the literature concerning the biocompatibility, the tolerability and the time resorption of these hemostatic agents are few [22,23,24,25,26]. Our study showed a longer resorption time than the data provided by the producers (7–14 days with regard to the oxidized and regenerated cellulose-based hemostatic; the failure rate for absorption is not clearly specified by the manufacturers). Therefore, it is important to keep in mind that failure or delay in the reabsorption of the hemostatic material can cause diagnostic mistakes that affect the patient’s therapeutic path. In fact, some of the patients had already undergone further diagnostic investigations before being enrolled in the study, such as scintigraphy and ultrasound-guided needle aspiration for residual material in the thyroid space in order to exclude a neoplastic recurrence. In particular, seven of these patients were subjected to strict ultrasound surveillance (three-month follow-up), one patient to MRI and three patients to scintigraphic evaluation that showed no areas of significant radiopertechnetate uptake and was negative for residual thyroid tissue in the anatomical loggia (Figure 5a–f). One of the last three patients had also undergone a needle aspiration evaluation that confirmed the presence of the hemostatic, described by the pathologist as “amorphous inorganic material referable to surgical sponge” (a similar case is described by Kurian E.M. et al., in which the patient, during follow-up after lung resection for carcinoma, underwent endobronchial ultrasound-guided transbronchial needle aspiration on suspicion of lymph node metastasis, but the cytological examination showed the presence of foreign material, later proven to be the Surgicel^®^ placed during surgery) [27,28]. Lastly, in two other patients, although they were not subjected to further diagnostic investigations, the presence of the swab was misinterpreted as post-surgical native thyroid residue. These examples make it clear how diagnostic mistakes cause unpleasant consequences for the patient’s diagnostic–therapeutic path and increase the clinical management cost.

The ultrasound appearance of oxidized regenerated cellulose swab has been described in some previous scientific papers [29,30]. Some authors have also described its radiological CT and MRI appearance, and even its cytology and histology [27,30,31,32]. In none of them, however, specific reference is made to the use of the oxidized regenerated material in thyroid surgery and its radiological appearance. Such studies describe the ultrasound appearance of the swab as a complex mass containing both hypoechoic and hyperechoic components, or as a formation with different ecogenic patterns (ipo-isoechoic with internal hyperechoic nodule, hypo-anechoic without internal hyperechoic nodule, completely anechoic) and/or different eco-structural characteristics (for example, capsulated or non-capsulated appearance) [29]. This wide variety of ultrasonographic aspects depends on the complex conformation that the pad assumes in the time after surgery, trapping, sub-solid, fluid and gaseous material within it parts (the latter being at the basis of the strongly hyperechoic interfaces that have been found in the follow-up of some patients.) [29]. In our experience, we found less heterogeneity in the findings, which showed a less variable and more easily recognizable appearance in the ultrasound.

The main key points to keep in mind during the ultrasound evaluation of patients undergoing thyroid surgery and treated with cellulose-based hemostatic in order to avoid diagnostic errors are as follows:Accurate clinical history collection. Knowing the type of surgical material used and the time elapsed since the surgery is crucial, since the residue persisted in 80% of the patients analyzed, with a greater prevalence within the first 1–2 years after surgery. It is therefore not rare in this type of examination to find positive but not necessarily pathological ultrasound findings.A careful multiple scanning US evaluation. Particular attention must be paid to the identification and recognition of the previously indicated patterns and margins evaluation, which is generally regular and never infiltrating.The Color Doppler evaluation, since it is an inorganic tissue without internal vascularization.The absence of concurrent pathological changes, such as cervical adenopathies or blood chemistry parameter alterations.

## 5. Conclusions

Topical hemostatic agents have proven to be a valuable aid to other hemostasis techniques in thyroid surgery. However, a deeper knowledge of reabsorption times and of the ultrasound features of these materials could reduce the number of diagnostic errors and the performance of inappropriate diagnostic tests. The preliminary data available show that the effectiveness hemostatic fibrin glue in preventing bleeding is comparable to oxidized regenerated cellulose swab; however, unlike the swab, the fibrin glue does not show any detectable residues during US examination.

## Figures and Tables

**Figure 1 cancers-15-02644-f001:**
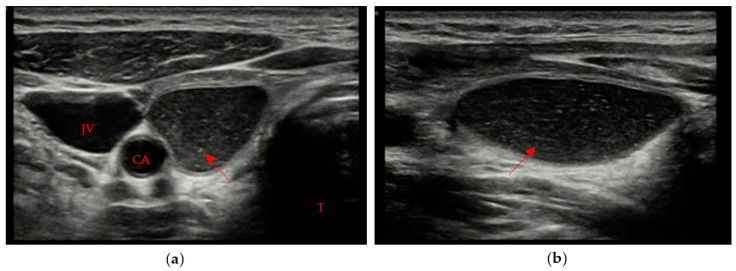
Residual swab pattern (red arrow) 1, characterized by a markedly hypoechoic pattern with an ecostructure of hyperechoic spots and well-defined margins; (**a**) Longitudinal view; (**b**) Axial view. Jugular vein (JV), Carotid artery (CA) and Trachea (T).

**Figure 2 cancers-15-02644-f002:**
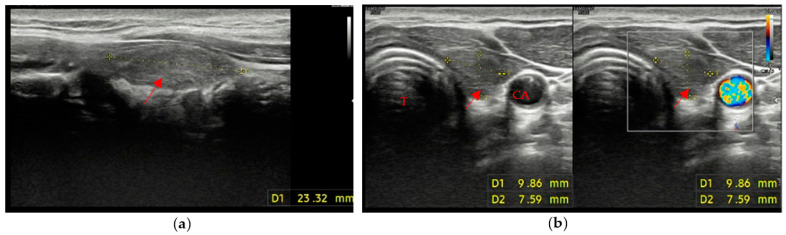
Residual swab pattern (red arrow) 2, characterized by a mildly hypoechoic/isoechoic ecostructure and ill-defined margins; (**a**) Longitudinal view; (**b**) Axial view. No vascular signal is seen at Color Doppler. Carotid artery (CA) and Trachea (T).

**Figure 3 cancers-15-02644-f003:**
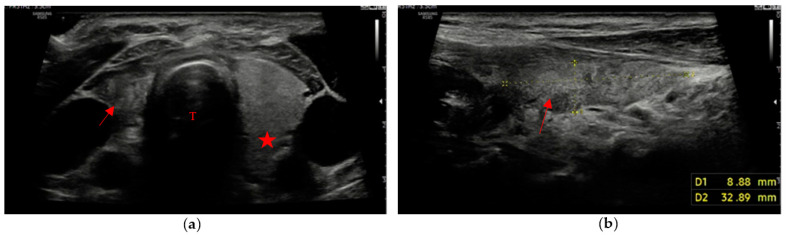
“Thyroid-like” shape of the swab (red arrow) in a right hemithyroidectomy (the native gland, pointed out by a red star, is still visible on the left). Trachea (T). (**a**) Axial view; (**b**) Longitudinal view.

**Figure 4 cancers-15-02644-f004:**
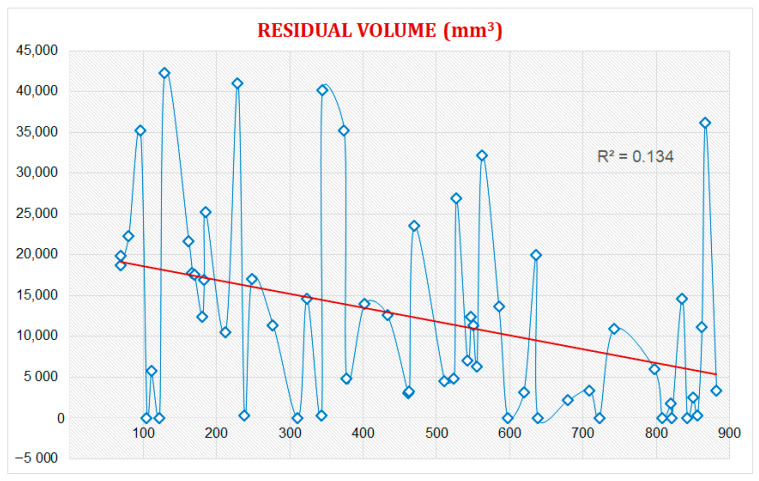
The residual volume of the swab, calculated in mm^3^, is reported on the ordinate axis; the swab was measured in the three axes, approximating its shape to that of an ellipsoid or the sum of other solid figures depending on the conformation for each individual case. The datum was set as a function of the time elapsed since the intervention (calculated in days and reported on the abscissa axis). The graph shows a large interindividual variability in patients and a weak and non-statistically significant negative correlation between volume and time (R^2^ = 0.134).

**Figure 5 cancers-15-02644-f005:**
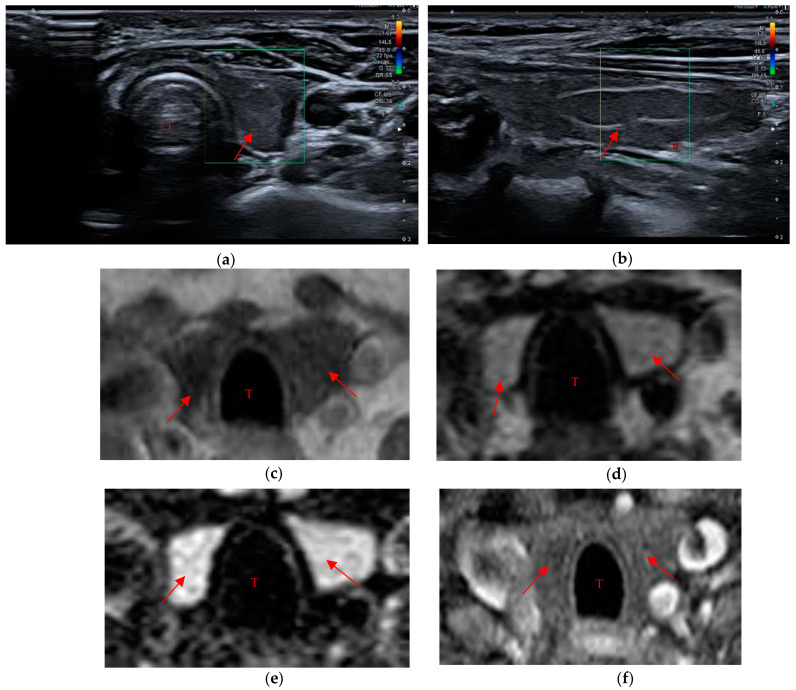
“Thyroid-like” shape of the residual swab (red arrow in the imagines) in a patient who underwent thyroidectomy. (**a**,**b**) The mildly hypoechoic pattern is recognized during ultrasound evaluation. (**c**–**e**) The swab in MR images appears moderately hypointense in T1w sequences and relatively hyperintense in T2 and in fat-suppressed T2. (**f**) No enhancement is seen in the post-contrast sequences, unlike what would occur in the native thyroid. Trachea (T).

**Table 1 cancers-15-02644-t001:** Demographic features.

Features	Total Population 84	Oxitamp^®^ 49	Tissel^®^ 35
Sex n (%)			
M	36 (42.9)	20 (40.8)	16 (45.7)
F	48 (57.1)	29 (59.2)	19 (54.3)
Age Mean ± DS (range)	56.2 ± 11.84 (20–87)	55.9 ± 12.17 (20–87)	56.7 ± 10.31 (33–80)
Surgery n (%)			
TT ^1^	77 (91.6)	44 (89.8)	33 (94.3)
LT ^2^	7 (8.4)	5 (10.2)	2 (5.7)
Histological diagnosis n (%)			
- Benign	45 (53.6)	26 (53.1)	19 (54.3)
- Malignant	39 (46.4)	23 (46.9)	16 (45.7)
Drainage volume			
1st post-operative day	43.59 ± 22.52 mL	45.19 ± 23.84 mL	35.94 ± 20.43 mL
2nd post-operative day	37.49 ± 21.87 mL	35.32 ± 23.26 mL	41.88 ± 20.48 ml
Hemostatic residue n (%)	39 (46.4%)	39 (79.59%)	0 (0%)
First evaluation Mean ± DS (range) mean months after surgery	8.03 ± 3.7 (3–24)	9.3 ± 4.2 (3–24)	6.25 ± 1.8 (3–9)

^1^ TT: Total Thyroidectomy; ^2^ LT: Lobectomy.

**Table 2 cancers-15-02644-t002:** Ultrasound features of oxidized regenerated cellulose in our experience.

Features	Pattern 1	Pattern 2
Echogenicity	Markedly hypoechoic with hyperechoic spots	Mildly hypoechoic/ isoechoic
Margins	Well-defined/ pseudo-capsulated	Ill-defined
Composition	Solid or almost completely solid ^1^	Solid or almost completely solid ^1^
Shape	Ovoidal	Thyroid-like
Color Doppler	No vascular signal	No vascular signal

^1^ No clearly fluid or gaseous areas were observed in the swab residues studied in the ultrasound.

## Data Availability

Data are not available due to privacy and ethics restrictions.

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
