# Peer review of "US Evaluation of Topical Hemostatic Agents in Post-Thyroidectomy"

_cancers, 2023, doi:10.3390/cancers15092644_

Round 1

Reviewer 1 Report

I read this manuscript with great interest. This is an original article that analyzes the surgical outcome of Oxitamp, a commonly used haemostatic, and compares the results with those obtained with a second haemostatic agent, Tissel (fibrin glue). For Surgeons involved in thyroid surgery it is not infrequent to encounter patients with a delay in the reabsorption of the haemostatic agent under examination, often mistaken for a recurrence of the disease, with an important effect on the patient's quality of life and an increase in legal disputes. Eighty-four patients were included in the work, in 49 the oxitamp was used as haemostatic, in 35 the surgeon had chosen the tissel for haemostasis at the end of the operation. All patients were evaluated with ultrasound within two years of surgery, and the size of the remnant found was correlated with time since surgery. The Authors conclude that the effectiveness in order to prevent bleeding of haemostatic fibrin-glue (Tissel) is comparable to oxidized regenerated cellulose swab (Oxitamp); however, unlike the swab, the fibrin-glue does not show any detectable residues at the US examination.

The sample size is adequate for the purpose of the study. The work is well structured and the data analysis well done and very interesting. The images shown in the work are clear and of great impact. References are adequate and reported correctly, according to the editorial rules of the Journal. Overall the work is well written.

In conclusion, I think that this manuscript is adequate for publication in this Journal, without the need for further revisions and can be accepted in current form.

Author Response

We thank reviewer 1for the attention dedicated to our manuscript and for the comments.

Reviewer 2 Report

This is a truly interting study, regaring the ultrasound features of hemostatic agents. A clinically relevant topic is accessed. 

However, there are some major issues in the manuscript that should be adressed.

The introduction can be shortended regaring its present content and should be complemented with some background that is now described in the results section. 

In the results section a proper discription of the time between surgery and ultrasound AND frequency of occurence of ultrasound features should be presented. 

The figures are of good quality but arrows naming the structures will make them much more informative.

In the discussion some patient specific results are presented for the first time (p7, line 223 and further), transfer them to the results section. 

In the discussion (p6, line 203) purpose of the study is advantage of one agent over the other. That is not true. Ultrasound features are studied, and that was the purpose. Furthermore, claims on no difference in reducing the volume of drained blood material and total blood losses, are not supported by any data. Those data should be added or these sentences should be removed (p6, line 205-210).

Author Response

We thank reviewer 2 for the attention dedicated to our manuscript and for the comments, which allowed us to improve our work.

This is a truly inserting study, regarding the ultrasound features of hemostatic agents. A clinically relevant topic is accessed. However, there are some major issues in the manuscript that should be addressed.

  1. The introduction can be shortened regarding its present content and should be complemented with some background that is now described in the results section.

The introduction has been modified and partially deepened. We have also moved some of the information previously contained in the results section, as suggested.

  1. In the results section a proper description of the time between surgery and ultrasound AND frequency of occurrence of ultrasound features should be presented. 

The relationship between time elapsed after surgery and the incidence of US findings has been examined in detail and deepened in the text.

  1. The figures are of good quality but arrows naming the structures will make them much more informative.

identifying marks have been added and described in the captions for all figures to make them much more informative as suggested.

  1. In the discussion some patient specific results are presented for the first time (p7, line 223 and further), transfer them to the results section. 

The text was modified according to the indications and an in-depth discussion was maintained, with the aim of commenting the relationship between our results and what was also reported in Literature.

  1. In the discussion (p6, line 203) purpose of the study is advantage of one agent over the other. That is not true. Ultrasound features are studied, and that was the purpose. Furthermore, claims on no difference in reducing the volume of drained blood material and total blood losses, are not supported by any data. Those data should be added or these sentences should be removed (p6, line 205-210).

A further section concerning differences in drained blood in the two groups of patients was added, accompanied by statistical analysis; the statistical analysis shows that no differences in the means of drained blood in the first two post-operative days between the two groups are present.

Reviewer 3 Report

The authors have written a very important paper documenting persistence of oxycellulose in the neck post thyroidetcomy. They document its persistence and provide images on US and MRI which inform the clinician that this can occur and how to differentiate from thyroid tissue and thyroid cancer recurrence. This is an important issue to delineate as I have had clinicians send back patients with this precise problem asking for surgical rexploration for recurrent thyroid cancer.

The comparison with tisseel is salient.

The paper is well written.

Author Response

We thank reviewer 3 for the attention dedicated to our manuscript and for the comments.

Reviewer 4 Report

The paper highlights an important issue when scanning this group of patients post-operatively. There are a few comments which should be addressed, which are detailed below.

Major Comments

-       Detail the statistical software used for the tests applied and the values used to determine significance in the methodology.

-          The results could be set out a little clearer. Consider the following points:

o   Provide a breakdown of the demographics of the patients, the number of patients having total or hemi-thyroidectomy, the number of follow scans, the average time the scans were performed following surgery. Given one of the aims of the paper was to compare differences in appearances between two different haemostatic materials it would be beneficial to compare these demographics between the groups to determine if the populations are similar. Were all the patients who had Tisseel scanned on average later than those who had Oxitamp? What were the differences in drained blood material, this was discussed in the discussion but not presented.

o   Adding the numbers of patients in your cohort with the US findings described in table 1 would be beneficial.

o   Is the amount of haemostatic material used for each patient the same volume – could this affect the time series regression and reabsorption? How many of the patients had multiple time point scans?

o   How was a significant reduction in size defined? What statistical test was used?

o   Was it the original report which described the findings of the Tisseel patients as having “fibrotic tissue following surgery" or was this the consensus from reviewing the images?

What is the quoted failure rate for absorption by the manufacturers?

Minor Comments

-         Who performed the US evaluation?

-        Try to avoid using the word “about” when describing a value e.g. line 113, give the value or use the term “approximately”.

-          Line starting 112 could be re-worded slightly e.g. “In 39 of these cases (approximately 80%) the presence of an ultrasonographically viewable residue was demonstrated…”

-          Table 2 should be described as a Figure.

Author Response

We thank reviewer 4 for the attention dedicated to our manuscript and for the comments, which allowed us to improve our work.

Major Comments

  1. Detail the statistical software used for the tests applied and the values used to determine significance in the methodology.

The software used was NCSS (Number Cruncher Statistical System) and Microsoft Excel. Information were added in the text.

  1. The results could be set out a little clearer. Consider the following points:

2.1.   Provide a breakdown of the demographics of the patients, the number of patients having total or hemi-thyroidectomy, the number of follow scans, the average time the scans were performed following surgery. Given one of the aims of the paper was to compare differences in appearances between two different haemostatic materials it would be beneficial to compare these demographics between the groups to determine if the populations are similar. Were all the patients who had Tisseel scanned on average later than those who had Oxitamp? What were the differences in drained blood material, this was discussed in the discussion but not presented.

All the requested information has been added in the text, and a table (table 1) summarizing the data has been provided. More information has been provided about the homogenization criteria adopted in the enrollment and selection phase. Furthermore, a further section concerning differences in drained blood in the two groups of patients was added, accompanied by statistical analysis.

2.2. Adding the numbers of patients in your cohort with the US findings described in table 1 would be beneficial.

The data was already present in the text as a percentage, it was also entered as a number.

2.3 Is the amount of haemostatic material used for each patient the same volume – could this affect the time series regression and reabsorption? How many of the patients had multiple time point scans?

Is not possible to trace the precise quantity of swab used for haemostasis, varying from patient to patient depending on the volume of the surgical cavity; the data have been reported into the text. Only 9 patients, as indicated in the text, had multiple time point scans.

2.4 How was a significant reduction in size defined? What statistical test was used?

A linear regression model was used. A statistically significant correlation between the time and volume variables is expressed by the value of the bivariate correlation coefficient R2 (as shown in figure 4).

2.5 Was it the original report which described the findings of the Tisseel patients as having “fibrotic tissue following surgery" or was this the consensus from reviewing the images?

It was described in the original report. Added in the text.

2.6 What is the quoted failure rate for absorption by the manufacturers?

 The failure rate for absorption is not clearly specified by the manufacturers. Added in the text.

Minor Comments

  1. Who performed the US evaluation?

All the scans were performed by a radiologist with more than twenty years of experience in ultrasound and in particular in the field of thyroid pathology. Added in the text.

  1. Try to avoid using the word “about” when describing a value e.g. line 113, give the value or use the term “approximately”.

Corrected in the text

  1. Line starting 112 could be re-worded slightly e.g. “In 39 of these cases (approximately 80%) the presence of an ultrasonographically viewable residue was demonstrated…”

Corrected in the text

  1. Table 2 should be described as a Figure.

Table 2 has been changed in figure 1 and all figures have been correctly numbered.

Round 2

Reviewer 2 Report

Overall I am satisfied with the changes applied to the manuscript in response to my comments.